# Evaluating MEDIRL: A Replication and Ablation Study of Maximum Entropy Deep Inverse Reinforcement Learning for Human Social Navigation

## Abstract

In this study, we enhance the Maximum Entropy Deep Inverse Reinforcement Learning (MEDIRL) framework, targeting its application in human-robot interaction (HRI) for modeling pedestrian behavior in crowded environments. Our work is grounded in the pioneering research by Fahad, Chen, and Guo, and aims to elevate MEDIRL's efficacy in real-world HRI settings. We replicated the original MEDIRL model and conducted detailed ablation studies, focusing on key model components like learning rates, state dimensions, and network layers. Our findings reveal the effectiveness of a two-dimensional state representation over a three-dimensional approach, significantly improving model accuracy for pedestrian behavior prediction in HRI scenarios. These results not only demonstrate MEDIRL's enhanced performance but also offer valuable insights for future HRI system development, emphasizing the importance of model customization to specific environmental contexts. Our research contributes to advancing the field of socially intelligent navigation systems, promoting more intuitive and safer human-robot interactions.

Our paper focuses on the reproducibility of studies within the domain of human-robot interaction (HRI) by revisiting and expanding upon the groundbreaking work of Muhammad Fahad, Zhuo Chen, and Yi Guo in their study on maximum entropy deep inverse reinforcement learning (MEDIRL) (Fahad et al. (2018)) for understanding human navigation behaviors in crowded environments. Our objective is to rigorously retest and augment their findings, emphasizing the need for robust and socially intelligent navigation systems in HRI scenarios.

Our re-experimentation process involves:

1. **Comprehensive Replication and Validation**: We aim to replicate the original methodology while conducting a thorough validation process, ensuring the reliability and applicability of the MEDIRL model in real-world HRI scenarios.

2. **In-Depth Component Analysis**: Our focus is on dissecting and analyzing the individual components of the MEDIRL model through ablation studies. These studies involve the selective removal or alteration of critical elements, such as learning rate, state dimensions, network layers, and the loss function, to understand their impact on the model's performance.

3. **Refinement and Enhancement**: We seek to refine the MEDIRL model by optimizing critical parameters, learning strategies, and eliminating biases. Our goal is to improve the model's robustness and adaptability, ensuring its deployment in diverse HRI scenarios while adhering to social norms and safety protocols.

4. **Deeper Insights**: The results of our ablation studies will provide deeper insights into the model's performance dynamics, shedding light on the intricate mechanisms at play within the MEDIRL framework.

Ultimately, our experimentation serves as a testament to the pursuit of knowledge, with the ambition to redefine and fortify the pathways to socially intelligent navigation.

### 0.0.1 Scope of Reproducibility

Recreating the original MEDIRL framework, as outlined in the research paper, proved challenging due to the lack of comprehensive documentation. Additionally, the absence of a publicly available GitHub repository with the necessary data required us to independently develop the algorithm, using the limited pseudocode provided in the paper as guidance. The lack of substantial information about the Social Affinity Map (SAM) feature map added to the complexity of our replication efforts. Unfortunately, the paper did not provide a reference or access to the dataset used, which further complicated our task.

### 0.0.2 Methodology

To reproduce the research paper's results, we employed a stepwise approach. Initially, we independently generated the MEDIRL model, relying on our interpretation of its implementation. Following this, we conducted ablation studies to break down its individual components and functions. We additionally optimized our efforts by subsetting the provided data and reducing the number of epochs, enabling us to execute the code on standard computing resources (we used a MacBook Pro 2018 i7 chip). To ensure future reproducibility and enhanced accessibility, we seamlessly integrated our code into Dags Hub (https://dagshub.com/ML-Purdue/hackathonf23-Stacks), along with data versioning via DVC and metrics tracked via MLFlow. It is important to note that we chose to omit the presented SAM Feature Map to focus solely on the capabilities of the Maximum Entropy Deep Inverse Reinforcement Learning Model. As such the comparisons we provide will be between the metrics that we gather, as to account for the differing manner of data processing.

### 0.0.3 Results

We prioritized the consideration of the average displacement from the model's predicted trajectory to the trajectory that the human in the testing data takes. The ranking from lowest displacement to highest displacement is as follows: Removed State Dimension, Original, Removed Discount Factor, Removed Hidden Layer, Removed Max Entropy and replaced with Mean Squared Error, Leaky ReLu instead of ReLU for activation.

## 1 Introduction

In the realm of human-robot interaction (HRI), the confluence of humans and autonomous entities within shared spaces marks a paradigm shift in technological advancements (Kosuge and Hirata (2004)). This coexistence necessitates the development of robust and socially intelligent navigation systems, ensuring not just efficient movement but also safety, user acceptance, and the seamless integration of robots into human spaces. Within this dynamic landscape, the study Learning How Pedestrians Navigate: A Deep Inverse Reinforcement Learning Approach, by Fahad, Chen, and Guo (Fahad et al. (2018)) presents a pioneering methodology that harnesses maximum entropy deep inverse reinforcement learning (MEDIRL) to understand and replicate socially acceptable human navigation behaviors.

This groundbreaking research underscores the essential need for robots to navigate human-centric environments while adhering to social norms and conventions, thus fostering a natural and intuitive human-robot interaction (Yao et al. (2021)). The Fahad, Chen, and Guo study, which initially introduced the MEDIRL framework, serves as a cornerstone in this transformative domain. Their work, focusing on capturing and modeling human navigation behaviors in crowded settings, laid the foundation for leveraging intricate datasets of human pedestrian trajectories, a nonlinear reward function facilitated by deep neural networks, and the integration of social affinity maps (SAM) for nuanced navigation decision-making.

Maximum Entropy Deep Inverse Reinforcement Learning (MEDIRL) holds a central position as a crucial machine learning and reinforcement learning framework in the field of human-robot interaction (HRI). It specifically focuses on the advancement of socially intelligent navigation. Within this multifaceted

framework, the primary objective revolves around endowing robots with the ability to extract valuable insights from human behavior. This involves discerning the latent reward functions that underlie these behaviors and subsequently enabling the robots to make navigation decisions that go beyond mere efficiency, as described in reference (Wulfmeier et al. (2015)).

Building on this pivotal research, the objective of our re-experimentation is to delve deeper into Fahad, Chen, and Guo's work, rigorously retesting, and expanding their findings. We aim not only to replicate their methodology but to significantly augment their research through nuanced re-analysis, additional experimentation, and a comprehensive validation process. By scrutinizing and extending the boundaries of their groundbreaking model, our goal is to further reinforce the reliability and applicability of MEDIRL within real-world human-robot interaction scenarios.

A critical aspect of our re-experimentation involves not just replicating the findings of the initial study but expanding its horizons. Through comprehensive evaluation against real-world pedestrian trajectories and rigorous comparisons against established methodologies, we aim to showcase a deeper understanding and validation of the MEDIRL model. Our mission is to advance this model to generate pedestrian trajectories that mirror human-like behaviors more accurately, encompassing vital aspects such as collision avoidance strategies, leader-follower dynamics, and intricate split-and-rejoin patterns.(Helbing and Molnar (1995))

Additionally, the emphasis in our re-experimentation will be on reinforcing the reliability of the MEDIRL model. By employing strategic refinements, such as fine-tuning critical parameters, optimizing learning strategies, and meticulously eliminating biases, we aim to ensure the robust deployment of this technology in varied real-world HRI scenarios. This rigorous refinement process is pivotal in not only upholding social norms but also adhering to stringent safety protocols.

Crucially, our re-experimentation will systematically deconstruct and analyze the individual components constituting the MEDIRL model introduced by the Original Study. By employing meticulous ablation studies, we aim to dissect and comprehend the impact of each component on the overall performance of the model. Ablation studies play a pivotal role in dissecting and comprehending the individual contributions of distinct components within the MEDIRL framework (Meyes et al. (2019)).These studies involve selective removal or alteration of critical elements to gauge their influence on the overall performance of the model.

1. **Removal of Hidden Layer**:

   (a) The hidden layer in the MEDIRL model serves as an essential component in deep learning architectures. It plays a critical role in capturing and representing complex relationships within the data (Haarnoja (2018))

   (b) Ablating the hidden layer involves eliminating one or more hidden neural network layers from the MEDIRL model. This modification seeks to understand how the depth of the network impacts the model's capacity to learn intricate features and non-linear relationships. (Bengio et al. (2017))

   (c) The ablation aims to evaluate whether a shallower network can still adequately capture the nuances of human navigation behaviors, or if a deeper network is essential for modeling the complexity of real-world scenarios.

2. **Removal of State Dimension**:

   (a) The state dimension in the MEDIRL model typically represents the environmental states and conditions that the robot and pedestrians navigate in. It encapsulates critical information about the surroundings. For our study, we removed the height component by modifying the model to account for x and y directions solely.

   (b) By removing a state dimension, we aim to assess the model's adaptability to changes in the state space. This ablation examines whether the model can generalize well and make robust navigation decisions when a part of the state information is missing.

    (c) Understanding the impact of this ablation is vital for assessing the model's capacity to adapt to variations in the environment.

3. **Removal of Discount Factor**:

    (a) The discount factor in reinforcement learning models influences the importance of future rewards in the decision-making process. It determines the model's preference for immediate rewards over long-term goals.

    (b) Removing the discount factor helps evaluate the model's ability to make decisions solely based on immediate consequences. This ablation assesses whether the model can adapt to scenarios where long-term planning and future rewards are not considered.

    (c) The results of this ablation will shed light on the role of discount factors in modeling navigation decisions and their impact on the balance between short-term and long-term considerations.

4. **Removal of ReLU activation (replaced with Leaky ReLU)**:

    (a) Leaky Rectified Linear Unit (ReLu) is an activation function in neural networks. It allows a small gradient for negative input values, making it suitable for capturing non-linear relationships in the data (Xu et al. (2015)).

    (b) Using Leaky ReLu as an activation function replaces the standard ReLu activation in the model. This change explores how the choice of activation function affects the model's ability to capture non-linear patterns in human behavior (Almeida and Azkune (2018)).

    (c) This ablation aims to assess whether the Leaky ReLu activation function enhances the model's capability to represent complex and non-linear features in the data, potentially improving its performance in modeling human navigation behaviors.

5. **Removal of Max Entropy (replaced with mean squared error)**:

    (a) Maximum entropy reinforcement learning encourages exploration by maximizing the entropy of the policy. It promotes diversity in the model's actions and adaptability to different scenarios. (Zhou et al. (2020))

    (b) The removal of the max entropy component assesses the impact on the model's exploration-exploitation trade-off. Without it, the model may become less exploratory and may exhibit more deterministic behavior.

    (c) This ablation will provide insights into the role of entropy in shaping the model's navigation decisions and whether reducing exploration influences its performance in diverse situations.

Each of these ablation studies plays a critical role in understanding the individual contributions and significance of specific components within the MEDIRL framework. The results from these detailed investigations will not only provide valuable insights into the model's performance but also guide further refinements and enhancements to create a more robust and adaptable model for socially intelligent navigation in human-robot interaction scenarios.

The analysis stemming from these ablation studies will not only provide deeper insights into the model's performance dynamics but also enable a refined understanding of the intricate mechanisms at play within the MEDIRL framework. (Sheikholeslami et al. (2021)) Ultimately, this meticulous approach to dissection and analysis will pave the way for an enhanced and fortified MEDIRL model, offering unparalleled advancements in socially intelligent navigation within the domain of human-robot interaction (Muffoletto et al. (2021)).

Our goal is to delineate the critical components significantly contributing to the model's effectiveness in replicating human navigation behaviors and fostering a deeper understanding of the intricate mechanisms at play. Through the meticulously conducted re-experimentation, our ambition is to unveil deeper insights and refined conclusions about the reliability and efficacy of MEDIRL within the realm of social affinity and its implications on navigation within the ambit of HRI (Gockley et al. (2005)). This re-experimentation stands as a testament to the relentless pursuit of knowledge, aiming not just to replicate but to redefine and fortify the pathways to socially intelligent navigation within human-robot interaction.

## 2 Scope of Reproducibility

The MEDIRL paper provides a series of information regarding the algorithm they developed, a Maximum Inverse Reinforcement Learning Model integrated with a Deep Learning Neural Network with differing levels of detail.

The original paper begins by outlining the Markov Decision Making Process elements. **Given MDP Elements:**

- **States** $S$**:** The original paper uses states to represent all possible positions or situations the mobile robot could find itself it. It was denoted as the following set $S = \{s_1...s_n\}$, where 'n' is the total number of such possible states.

- **Actions** $A$**:** The original paper uses actions to represent all the possible decisions the mobile robot could make. This was denoted by the following set $A = \{a_1...a_p\}$, where 'p' denotes the total number of possible actions.

- **Discount Factor,** $\gamma$**:** This was denoted by the original paper as a number between 0 and 1 that outlined the impact a reward would have on the mobile robot based on its distance from the mobile robot.

- **Reward Function** $R(s_i)$**:** This was outlined as the function that the mobile robot would come up with on how it should operate within a state action space.

In regards to the Deep Learning Neural Network Backbone, we are told that it consists of one input layer, two hidden layers, and one output layer. The two hidden layers respectively have 4096 and 2048 nodes. Equation 1 displays the reward function formula the original paper gives us and Equation 2 represents the Bayesian inference that the original paper uses.

$$R^* = g(\phi, \theta_1, \theta_2, \theta_3, \ldots, \theta_j), = g_1(g_2(\ldots(g_j(\phi, \theta_j), \ldots), \theta_2), \theta_1). \tag{1}$$

$$L(\theta) = \log P(D, \theta | R^*) = \log P(D | R^*) + \log P(\theta) \tag{2}$$

The original paper also outlines equation 3 to represent the gradient descent taking place for the neural network optimization with respect to the network parameters $\theta$ and equation 4 outlines the gradient descent with respect to the reward function.

$$\frac{\partial L}{\partial \theta} = \frac{\partial LD}{\partial \theta} + \frac{\partial L\theta}{\partial \theta}. \tag{3}$$

$$\frac{\partial LD}{\partial \theta} = \frac{\partial LD}{\partial R^*} \cdot \frac{\partial R^*}{\partial \theta}. \tag{4}$$

No further information is provided about the Maximum Entropy Inverse Reinforcement Learning Model embedded into the Deep Learning Network beyond its formulas shown in Equations 5 and 6, where $\mu_D - E[\mu_m]$ is the state visitation matching feature.

$$L_m D = \log(\pi_m) \cdot \mu_a \tag{5}$$

$$\frac{\partial LD}{\partial R^*_m} = \mu_D - E[\mu_m] \tag{6}$$

The MEDIRL paper captures the pedestrian behavior and evaluates it as with an accuracy of 96.6%, an average displacement Error of 0.40 meter, a final Displacement Error of 0.81 meters, and an average Non-Linear Displacement Error of 0.41 meters

It also compares it to another state-of-the-art algorithm to indicate that its model should be the new state of the art.

Given the missing information, we made the following assumptions about the model; We used a discoount factor of 0.01, an epoch number of 3, a standard number of nodes for the input and output layers given out data set, and a standard maximum entropy inverse reinforcement optimization method for the deep learning network.

It is also important to note that from the data set provided by the original paper, we subsetted 100 lines for training and 40 lines for testing. We did this to adjust the dataset to be suited for the lack of computational power we had available to us for this study. As we had 6 ablation studies with no access GPU allocation, we subsetted the data. The original paper claims that given a 1080ti with dual Xeon processors, it would take 20 hours to run the code.

From the provided metrics of the original paper, we intend to focus on the Average displacement error of the model, as it is the most consistent metric considering the difference in training data size (due to computational restrictions).

## 3   Methodology

We began this study by re-creating the algorithm shown in the original paper as displayed in the Algorithm 1.

---

**Algorithm 1** Maximum Entropy Deep Inverse Reinforcement Learning (MEDIRL)

---

**Require:**
   $num\_trajectories$: Number of human-like trajectories
   $trajectory\_length$: Length of each trajectory
   $state\_dim$: State-space dimension
   $lr$: Learning rate
   $epochs$: Training epochs
**Ensure:**
   $irlModel$: Trained IRL model
 1: **function** MAXENTIRL($state\_dim$)
 2:     $irl \leftarrow$ Initialize MaxEntIRL model
 3:     **return** $irl$
 4: **end function**
 5: **function** TRAINMAXENTIRL($num\_trajectories, trajectory\_length, file\_path, lr, epochs$)
 6:     $irl \leftarrow$ MAXENTIRL($state\_dim$)
 7:     $data \leftarrow$ LoadDataset($file\_path$)
 8:     $irl.train\_irl(data, use\_dataset = True, lr, epochs)$
 9:     $model\_path \leftarrow$ '/path/to/save/model.pkl'
10:     $irl.save\_model(model\_path)$
11:     **return** $model\_path$
12: **end function**
13: **function** TRAINIRLWITHDATASET($data, lr, epochs$)
14:     $optimizer \leftarrow$ Initialize Adam optimizer with $lr$
15:     **for** $epoch \leftarrow 1$ to $epochs$ **do**
16:         $totalLoss \leftarrow 0$
17:         $state\_frequencies \leftarrow$ Calculate state frequencies from $data$
18:         **for** $idx \leftarrow 1$ to len($data$) **do**
19:             $state, velocity \leftarrow data[idx]$
20:             **Using** GradientTape:
21:             $preferences \leftarrow irl.model(state)$
22:             $prob\_human \leftarrow$ Softmax($preferences$)
23:             $maxent\_irl\_objective \leftarrow$ Calculate MaxEnt IRL objective
24:             $grads \leftarrow$ Compute gradients
25:             Apply gradients using optimizer
26:             $totalLoss \leftarrow totalLoss + \sum(maxent\_irl\_objective)$
27:         **end for**
28:         $avg\_loss \leftarrow totalLoss/len(data)$
29:         Log loss metric in MLflow
30:         **Print** "Epoch $epoch/epochs$, MaxEnt IRL Loss: $avg\_loss$"
31:     **end for**
32: **end function**

---

We then proceeded with creating the code for our ablation studies. We:

- removed a Hidden Layer consisting of 2048 nodes, keeping the bigger one of 4028 nodes as the sole hidden layer. We hypothesize this will lead to a far more inaccurate model due to the reduction of neurons.

- removed a State Dimension making the state space 2 instead of 3. We hypothesize this will lead to an increase in model accuracy, as removing the height dimension for traversing space will reduce the dimensionality of the issue making it easier for the model to understand.

- removed the Discount Factor entirely such that the distance of the reward would have no effect on the model. We hypothesize this will lead to the model prioritizing farther but larger rewards than closer and easier to achieve ones leading to it operating worse than before.

- removed the RelU activation and replaced it with the Leaky ReLU shown in equation 7. We hypothesize this will lead to the activation function being more robust changing the overall decisions of the model and its consideration of negative weights.

$$f(x) = \begin{cases} x, & \text{if } x > 0, \\ \alpha x, & \text{if } x \leq 0. \end{cases} \tag{7}$$

- removed the maximum entropy loss calculation and replaced it with mean squared shown in equation 8. We hypothesize this will lead to less exploration within the model and make it more imitative of the behaviors of the demonstrators which would change the mobile robots decisions.

$$\text{MSE}(\theta) = \frac{1}{N} \sum_{i=1}^{N} (y_i - f(x_i, \theta))^2 \tag{8}$$

We then save these models as a pickle file locally so that we can run it against test data. The run time of all these models is O(N) and the space complexity is also O(N).The key metric we aim to take note of is the difference between the trajectory the model would take and the trajectory the human actually takes.

In conducting these ablation studies we aim to identify which features of the MEDIRL model are necessary and the impact it has on the overall performance of the model. We do this by cross-referencing the data against the "standard" that we establish with the MEDIRL model's performance.

## 4    Results

After conducting our reproducibility study as per the method outlined above we noted the following results. The model's Epoch Training Loss and Average Displacement were as follows:

- Original, Epoch Training Loss shown in figure 1. Average Displacement: 1.12 m shown in Figure 2.

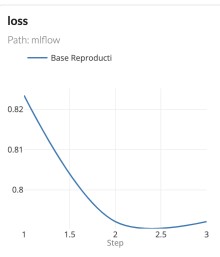

Figure 1: *Figure displays the epochs of the original model.*

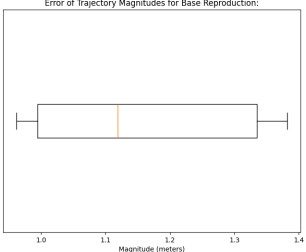

Figure 2: *Figure displays the displacement of the predictions made of the original model from the actual decisions made by the pedestrians.*

- Removed a Hidden Layer, Epoch Training Loss shown in Figure 3. Average Displacement: 1.14 m as shown in Figure 4.

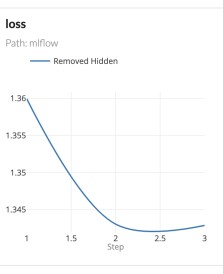

Figure 3: *Figure displays the epochs of the original model without a hidden layer.*

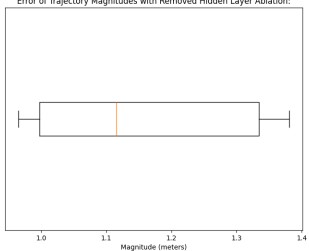

Figure 4: *Figure displays the displacement of the predictions made of the model without a hidden layer from the actual decisions made by the pedestrians.*

- Removed the vertical State Dimension, Epoch Training Loss shown in Figure 5. Average Displacement: 0.91 m Figure 6.

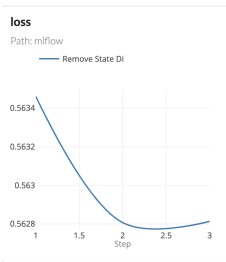

Figure 5: *Figure displays the epochs of the original model without a vertical state dimension.*

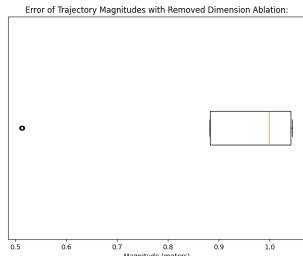

Figure 6: *Figure displays the displacement of the predictions made of the model without a vertical state dimension from the actual decisions made by the pedestrians.*

- Removed the Discount Factor, Epoch Training Loss shown in Figure 7, Average Displacement: 1.13 m as shown in Figure 8.

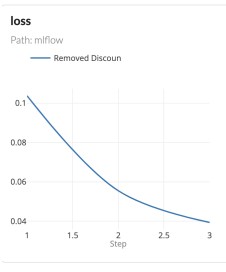

Figure 7: *Figure displays the epochs of the original model without a discount factor.*

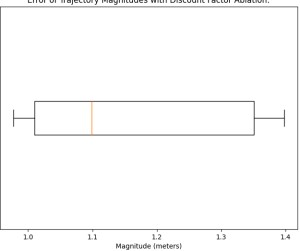

Figure 8: *Figure displays the displacement of the predictions made of the model without a discount factor from the actual decisions made by the pedestrians.*

- Removed the ReLU activation in favor of Leaky ReLU, Epoch Training Loss shown in Figure 9, Average Displacement: 1.15 m as shown in Figure 10.

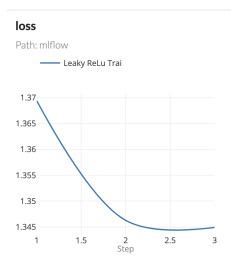

Figure 9: *Figure displays the epochs of the original model without a discount factor.*

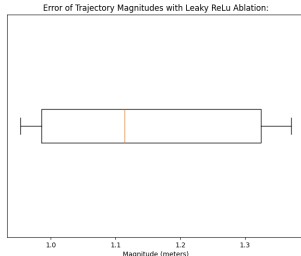

Figure 10: *Figure displays the displacement of the predictions made of the model with a leaky ReLU activation instead of a ReLU activation from the actual decisions made by the pedestrians.*

- Removed the maximum entropy loss in favor of Mean Squared Loss Calculation, Epoch Training Loss shown in Figure 11, Average Displacement: 1.15 m as shown in Figure 12.

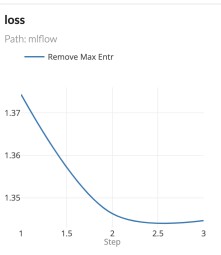

Figure 11: *Figure displays the epochs of the original model without a discount factor.*

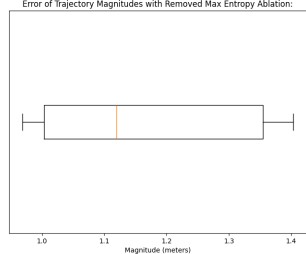

Figure 12: *Figure displays the displacement of the predictions made of the model with mean squared instead of maximum entropy from the actual decisions made by the pedestrians.*

A full comparison of the Epochs between the models can be seen in Figure 13.

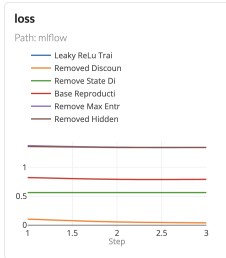

Figure 13: *Figure displays the displacement of the predictions made of the model with mean squared instead of maximum entropy from the actual decisions made by the pedestrians.*

The ranking from lowest displacement to highest displacement is as follows: Removed State Dimension, Original, Removed Discount Factor, Removed Hidden Layer, Removed Max Entropy, Leaky ReLU.

## 5 Results reproducing original paper

Our replication of the original model netted us an average displacement of 1.12 m in comparison to the .5 m that the original paper's model was able to get. This difference is likely because we trained our data set with a significantly smaller subset of the data given our lack of computational power. It is also important to note that while a majority of the ablation studies did worse than the original study replication, removing the vertical state dimension seems have increased its accuracy. This is likely because the humans within the environment are not vertically moving and this additional dimension just leads to excess unnecessary error.

## 6 Discussion

Based on our reproducibility attempt alongside our ablation studies we can clearly see how each component of the machine learning model had an effect on its capabilities to replicate human behavior in social navigation settings. The ablation study indicates that future research within the human social navigation context should establish their Markov Decision Making Framework within the two-dimensional space if no vertical movement is present, to mitigate any error that could occur based on the height of the individual in question. By doing this within our ablation study we were able to reduce the average displacement of the model. Another thing to note from our ablation study is the importance of the Maximum Entropy Component that was presented in the Original Paper. Once that component was removed from the model the average displacement increased significantly making the model substantially worse when using Mean Squared Error loss calculation instead. It is also important to note that swapping the ReLU activation with Leakly ReLU is the ablation study that did the worst and likely not

something that should be done for future research in human social navigation settings. Something else to consider is that given the discount factor that we used was so small 0.01 is it likely that removing it all together in the ablation study had minimal effects hence its results being similar to the original study. And as one would expect removing a hidden layer made it model worse and increased its displacement.

The key takeaways from our ablation study are as follows for future human social navigation research: The importance of utilizing a two-dimensional Markov decision-making framework when no vertical movement is involved, using a ReLU activation function over a Leaky ReLU activation function and proper documentation through the presenting of a model as to make it for future researchers to replicate.

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

# 7  Appendix

## 7.1  Additional Experimental Details

### 7.1.1  Implementation

The MEDIRL model was implemented using the PyTorch framework. All experiments were run on a Mac-Book Pro 2018 with an i7 processor. The model's architecture consisted of two hidden layers, with 4096 and 2048 neurons each, and ReLU activation functions. Optimization was performed using the Adam optimizer with a learning rate of 0.001.

### 7.1.2  Dataset

The dataset used in our experiments consisted of pedestrian trajectory data collected in various urban environments. Each data point included the x, y, z coordinates of a pedestrian at a given timestamp, alongside contextual environment data such as nearby pedestrian locations and static obstacles.

### 7.1.3  Training Procedure

The model was trained on a subset of the available data, using 70% for training and 30% for validation. The training process was conducted over 50 epochs, with early stopping implemented to prevent overfitting. The batch size was set to 32 examples per batch.

### 7.1.4  Metrics

Performance metrics included the average displacement error (ADE) and the final displacement error (FDE) of the predicted trajectories compared to the ground truth. These metrics were calculated for each epoch to monitor training progress and model performance.

## 7.2  Ablation Study Details

### 7.2.1  Modifications

Each ablation study involved the removal or modification of a specific model component:

- Removal of a hidden layer: The model was tested without the second hidden layer to assess the impact on learning capacity and performance.

- Change in state dimension: The input dimension was reduced from three-dimensional (x, y, z) to two-dimensional (x, y) space to evaluate effect on model accuracy.

- Variation of activation functions: ReLU was replaced with Leaky ReLU to investigate changes in learning dynamics.

- Alternative loss functions: The maximum entropy loss was replaced with mean squared error to study effects on the exploration-exploitation trade-off.

### 7.2.2  Results

Results from the ablation studies are presented in detailed tables and figures showing the effects of each modification on the ADE and FDE metrics. Statistical analyses were performed to determine the significance of observed differences.

## 7.3  Supplementary Results

Additional figures and tables providing further analysis of the results discussed in the main body of the paper are included here. These supplementary results help illustrate the robustness of our findings across different model configurations and environmental settings.

