# OpenReview forum: "Evaluating MEDIRL: A Replication and Ablation Study of Maximum Entropy Deep Inverse Reinforcement Learning for Human Social Navigation"
_TMLR — Rejected by TMLR_

### Review · Reviewer_wSpG · 2024-06-29

**Summary Of Contributions:**

The paper aims to reproduce and conduct experiments following the prior work "Learning how pedestrians navigate: A deep inverse reinforcement learning approach" [Fahad et al. IROS 2018].  The prior work addressed the problem of trajectory prediction of pedestrians (i.e. what actions a pedestrian in a multi-agent environment with obstacles will take to navigate from an initial position to a target position) and introduced maximum entropy deep inverse reinforcement learning (MEDIRL) for learning a reward function that can be then used to learn a pedestrian movement policy using approximate value iteration.  In the original MEDIRL paper, the ATC dataset (https://dil.atr.jp/crest2010_HRI/ATC_dataset/) was used, with 10M episodes used for training and 1500 randomly sampled pedestrian trajectories used for evaluation.

The submitted manuscript takes the MEDIRL model and investigates how changes to the model affects its performance.  Several ablations / model modifications are applied, including removal of hidden layers, removal of height component in state dimensions, removal of the discount factor, replacement of RELU with leaky ReLU, replacement max entropy loss with MSE loss.  Due to resource constraints, the model variants are trained and evaluated on a subset of the original data (100 for train, 40 for test).  For evaluation, the average displacement error (error between the predicted and actual measured pedestrian trajectories from the dataset) is reported.

**Audience:**

No

**Broader Impact Concerns:**

No broader impact concerns is included.  A statement of how the model can be used and potential pitfalls can be useful.

**Claims And Evidence:**

No

**Requested Changes:**

Significant re-work and additional experiments are required before this work is of potential interest to the community
- Experiments should be done on a larger scale with GPUs.  There should also be consideration of whether there are more interesting model variations to explore, for instance different architectures for predicting the reward.
- Experiments should include the different metrics from prior work ("Average Non-Linear Displacement Error" and "Final Displacement Error" should be included in addition to "Average Displacement Error")
- Related work need to be properly discussed and compared against.  Experiments should include comparison against recent work.
- There should be concise tables that compare different model variants.
- Paper will need to be rewritten to provide a improved overview of task and MEDIRL and reduce description of basic concepts like ReLU
  - More information should be provided on the task that is being addressed.  What is the goal of the task?
  - A better overview of MEDIRL should be provided.  What are the key elements of MEDIRL?  Why is MEDIRL a work that is worth reproducing and interesting to the community?   Why is it "groundbreaking"?
  - The first part (section 0) should be removed with relevant information moved into appropriate sections
  - Summary tables should be provided to concisely compare the different models
    - There is no need to show the loss curve or to show separate figures of displacement.  These figures are hard to interpret.
- The dataset used should be described and cited.  The original MEDIRL used:
    - Person tracking in large public spaces using 3-D range sensors [Brščić et al. 2013] (https://dil.atr.jp/crest2010_HRI/ATC_dataset/)
- Claims need to be toned down

**Strengths And Weaknesses:**

Strengths
- There is an attempt to reproduce MEDIRL and conduct ablation studies

Weaknesses
- There is limited contribution as the work fails to properly reproduce MEDIRL experiments
  - The dataset used for training and evaluation is much smaller than the dataset used in the original paper
  - The ablations that are conducted does not provide any insight into possible improvements in model design
  - Not all metrics used in MEDIRL are reported
- Limited discussion of recent prior work on pedestrian trajectory prediction.  It is unclear whether a re-investigation of the MEDIRL model is useful to the community or whether newer work has would be more relevant to investigate.
  - The manuscript also did not cite or describe the original dataset used for MEDIRL
- The writing and presentation is poor
  - It was difficult to understand the problem setting and the original MEDIRL model from the manuscript
  - The experimental results are not clearly summarized in a table allowing for easy comparison
  - The writing was also very verbose, with too much space/words dedicated to the description of the ablation conditions

---

### Review · Reviewer_Hc9e · 2024-07-14

**Summary Of Contributions:**

This paper replicates the Maximum Entropy Deep Inverse Reinforcement Learning (MEDIRL) model originally introduced by Fahad, Chen, and Guo (2018). It aims to find key model components that affect model performance and validate the original findings and ensure the robustness of the MEDIRL framework in modeling human behavior. For the ablation studies, the authors explored performing ablation study on individual components of the MEDIRL framework, including removing hidden layers, removing one state dimension (the height component), removing the discount factor, and replacing ReLU with Leaky ReLU, and removing the max entropy term from the loss function and replacing with mean squared error. It finds that the reproduced result is worse than the original study by Fahad, Chen, and Guo (2018), it also finds that removing the height component from its state information seems to have increased its accuracy. It reveals the importance of the max entropy component is vital and removing it causes the average displacement error to increase significantly. LeakyReLU also makes the model worse than using ReLU. Removing a hidden layer also made the model worse.

**Audience:**

No

**Broader Impact Concerns:**

This paper can have impact in the social navigation community with IRL. There is no ethical concern.

**Claims And Evidence:**

No

**Requested Changes:**

1. Clearly states the experimental details, including the dataset size, how it is collected, and maybe some example data samples.
2. Improve paper writing. Use appropriate and formal mathematical notations. When saying things like remove discount factor, clearly state what it means (does it mean set gamma=0 or gamma=1?). Replace algorithm 1 the language expressions with math notations for easier reading and understanding.
3. improve the figures and results part. The figures are hard to read, and the captions need to be improved to reflect what has been shown and concluded from the figures.

**Strengths And Weaknesses:**

Strength: the paper did ablation study to show that the height component is not vital for modeling human navigation behavior. It also shows through ablation study that it is important to keep hidden layers, keep ReLU, and keep the max entropy term.

Weakness: there are aspects where the paper lacks:
1. The conclusions are not surprising. Intuitively, it is easy to understand that the height component is not vital for human social navigation, and other conclusions such as remove hidden layer, and remove max entropy term makes the model worse, are also well studied and did not bring new insights to the community.
2. The writing and experiment sections are poorly written. There are several typos or inappropriate expressions in both the text section and equations. For example, the equation (3) (4) and (6) uses the term "LD" "L\theta" to represent gradient, which is not standard way to write these in the literature. On page 6, paragraph 2, line 1, the last word "discoount" should be "discount". The experiments are not described in clear details, such as what dataset is used here, and how large is the dataset, and if the data is selected from the original reference paper, how that selection is being done. The result figures are made with small fonts and confusing labels and legends, which makes it hard to understand the conclusion.
3. The results are not convincing. Even though the authors presented with several findings, since the authors claim their results are worse than in the original paper by Fahad, Chen, and Guo (2018), it is hard to draw convincing conclusions from the study. As the authors claim their training dataset is much smaller dataset, but a smaller dataset could lead to overfitting and thus it is not clear why the performance is even worse instead of better? Thus, the findings obtained with a very small dataset may not be generalizable to other human social navigation scenarios.

---

### Review · Reviewer_MN3t · 2024-08-27

**Summary Of Contributions:**

The paper aims to reproduce the results of " Learning How Pedestrians Navigate: A Deep Inverse Reinforcement Learning Approach", a 2018 paper by Fahad, Chen, and Guo. This paper uses a maximum entropy inverse RL approach to model pedestrian navigation, using a social affinity map (SAM) as an input feature to model motion affinity of nearby individuals. This reproduction omits the SAM feature, runs for fewer epochs, and uses a subset of the initial pedestrian data to reduce computational needs.

The method is then further ablated in the following ways:

* The hidden layer of the neural net is removed / kept.
* The state position of objects is changed to just (x,y) rather than (x,y,height)
* The discount factor in RL is removed / kept (changed to a value of 1 or not)
* The ReLU activation is changed to Leaky ReLU or not
* The max-ent loss is removed / kept (when removed, a squared error loss is used instead)

**Audience:**

No

**Claims And Evidence:**

Yes

**Requested Changes:**

There are not single changes that would affect my opinion of the work. The issues within the paper are fairly extensive and systematic.

**Strengths And Weaknesses:**

The paper is clear on the ablations run for this paper, but has a significant number of weaknesses.

First, by making many changes for the sake of efficiency (reducing dataset size), the experiment setup can no longer be a 1:1 reproduction of the original paper. As noted, they find a displacement error of 1.12m compared to 0.5m from the original paper, but there's no way to really compare these numbers.

A number of citations are also just strange, for example a citation on the importance of the hidden layer in deep learning is cited to the Haarnoja 2018 paper about SAC. This isn't a great citation because the SAC paper is not about hidden layers, it's about soft actor critic. It would be better to either leave this uncited or use a more generic source discussing neural net architectures.

There is a lot of padding, with sentences like "Ultimately, our experimentation serves as a testament to the pursuit of knowledge, with the ambition to redefine and fortify the pathways to socially intelligent navigation."

The authors link their code implementation in Dags Hub, which would normally be good, but breaks double blind reviewing because the linked repo does not redact the author names. (I stopped reading this link once I realized what was happening.)

The figures in the paper are hard to read, and could have larger text.

And overall, the ablations just aren't that interesting. They are very specific to this problem setting, focused almost exclusively on model architecture fitting rather than social navigation, and therefore not giving many useful takeaways. To quote from the TMLR acceptance criteria:

"Here's an example on how to use the criteria above. A machine learning class report that re-runs the experiments of a published paper has educational value to the students involved. But if it doesn't surface generalizable insights, it is unlikely to be of interest to (even a subset of) the TMLR audience, and so could be rejected based on this criterion. On the other hand, a proper reproducibility report that systematically studies the robustness or generalizability of a published method and lays out actionable lessons for its audience could satisfy this criterion."

I do not think this paper surfaces generalizable insights, nor does it study the robustness of the original paper, due to making many changes to the original paper's setup that prevents direct comparison. The claimed general lessons like "leaky ReLU did worse" do not feel strong enough to me. The choice of changes do not feel particularly motivated - they feel more like ablations that were run for the sake of running them.

---

### Decision · Action_Editor_Zwev · 2024-10-05

**Recommendation:** Reject

**Comment:**

There is a consensus on the rejection of the paper.

wSpG:"The submission does not provide an adequate replication of Maximum Entropy Deep Inverse Reinforcement Learning (MEDIRL), with weak ablations and poor writing, and will not be of interest to the TMLR community."

**Audience:**

The intended audience is research community working on human-robot interaction (HRI) and, more specifically, socially intelligent navigation systems, although in its current form that is challenging.

**Claims And Evidence:**

This paper investigates the reproducibility and effectiveness of Maximum Entropy Deep Inverse Reinforcement Learning (MEDIRL) for modeling human social navigation in crowded environments. The authors specifically focus on replicating and expanding upon a 2018 study by Fahad, Chen, and Guo that used MEDIRL to predict pedestrian trajectories.

While acknowledging the paper's attempt to reproduce and analyze the MEDIRL model, the reviewers identify several significant weaknesses that limit the paper's impact and generalizability.

Overall, the reviewers agree that the paper in its current form is not suitable for publication. They suggest substantial revisions, including:
1) Conducting experiments on a larger scale with sufficient computational resources. 2) Providing a more comprehensive and detailed description of the original MEDIRL model and the dataset used. 3) Designing more insightful ablation studies that explore a wider range of model variations. 4) Performing robust statistical analyses to support the findings. 5) Significantly improving the writing quality, clarity, and presentation of the results. 5) The reviewers also emphasize the importance of comparing the MEDIRL model with more recent work in pedestrian trajectory prediction to demonstrate its relevance and potential contributions to the field.